# Mechanistic Anomaly Detection for "Quirky" Language Models

**David O. Johnston** *
Eleuther AI
Washington, DC 20010, USA
`{david}@eleuther.ai`

**Arkajyoti Chakraborty**
University of California Santa Cruz
California, 95054, USA

**Nora Belrose**
Eleuther AI
Washington, DC 20010, USA

## Abstract

As LLMs grow in capability, the task of supervising LLMs becomes more challenging. Supervision failures can occur if LLMs are sensitive to factors that supervisors are unaware of. We investigate **Mechanistic Anomaly Detection** (MAD) as a technique to augment supervision of capable models; we use internal model features to identify anomalous training signals so they can be investigated or discarded. We train detectors to flag points from the test environment that differ substantially from the training environment, and experiment with a large variety of detector features and scoring rules to detect anomalies in a set of "quirky" language models. We find that detectors can achieve high discrimination on some tasks, but no detector is effective across all models and tasks. MAD techniques may be effective in low-stakes applications, but advances in both detection and evaluation are likely needed if they are to be used in high stakes settings.

## 1 Introduction

Large language models (LLMs) excel at a wide range of tasks, yet as their capabilities grow, ensuring effective human oversight becomes increasingly challenging. In many cases, LLMs may internalize and act on complex cues that are opaque to supervisors,[1] risking the reinforcement of superficially desirable but fundamentally flawed behavior.

These considerations have motivated **Mechanistic Anomaly Detection (MAD)**: instead of attempting to understand all facets of a model's internal reasoning, MAD focuses on extracting a signal that indicates when an episode is anomalous—flagging behaviors that deviate from the expected patterns without assuming specific trigger mechanisms (Christiano, 2022). This signal could be used to filter out anomalous episodes, or to trigger deeper investigation.

We evaluate MAD on intentionally modified datasets which we call "quirky datasets", first introduced by Mallen et al. (2024). These datasets simulate an oversight challenge: models are trained to exhibit both normal and anomalous behaviour, anomaly detectors are then trained on easy, correctly labeled examples and we test how well these detectors generalize to more difficult scenarios where anomalies arise. (Further details on the dataset and experimental setup are provided in Section 3.1.)

Our three main findings are:

- **Model-dependent detection:** The performance of anomaly detectors varies significantly between models.

---

*David and Nora are funded by a grant from Open Philanthropy.

[1]For example: adversarial examples in image models exploit previously unknown regularities of different image classes (Ilyas et al., 2019) and current frontier LLMs are known to acquire sycophancy biases from behaviour fine-tuning (Malmqvist, 2024), which may arise because models respond to the way supervisors are affected by flattery even though supervisors are unaware of this themselves.

- **Task specificity:** Detection efficacy is strongly task-dependent; for example, arithmetic tasks generally yield higher discriminative performance.
- **Correlation with behavior strength:** Larger deviations in behavior generally make anomalies easier to detect, though this is not universally true.

These are exploratory experiments testing the MAD concept. While we induce a number of different kinds of anomalous behaviour in our quirky datasets, they remain datasets of relatively simple tasks with simple triggers for anomalous behaviour. On these datasets, we find that anomaly detectors may sometimes be highly discriminative of anomalous inputs, but overall results are inconsistent. Our evidence indicates that simple anomaly detection methods may generally work as weak detectors, but questions remain about how reliably they will work in realistic settings.

## 2 RELATED WORK

Anomaly detection is closely related to *backdoor detection*. A backdoored language model is a model that has been trained to respond maliciously to texts with certain triggers in them – thus the model responds to cues the supervisor is unaware of (i.e. the backdoor triggers), and these cues cause anomalous behaviour. We could apply anomaly detection techniques to attempt to find backdoored behaviour at inference time.[2] The key difference between anomaly detection and backdoor detection is that anomaly detection avoids making assumptions about how anomalous behaviour might arise – it may not be due to rare codewords or patterns in the input, which is typically assumed for backdoor detection.

Our work differs in setting and methods from existing runtime backdoor detection approaches. In contrast to Gao et al. (2019); Guan et al. (2025), our models exhibit input-dependent behaviour in both normal and quirky operation modes (plus we focus on text rather than images). Our "activations/Mahalanobis" detector is a version of the "DeepFeatures" detector of Subedar et al. (2019). Unlike ours, their work focuses on image models, and we evaluate many other types of detector. We assume that external sources of ground truth are unavailable, unlike Mo et al. (2023). Even though Hayase et al. (2021) assumes that anomalous examples are rare and that no trusted dataset is available – both assumptions that are flipped in our paper – we still experiment with a modified version of the detector they introduce, which we call *quantum entropy*.

We hypothesize that anomalous examples produce network behaviours that are statistical outliers with respect to the distribution of normal network behaviours. Many of our methods are drawn from the field of outlier detection. (Wang et al., 2019). Bai et al. (2022) used outlier detection during fine tuning to reject harmful or unusual requests during LLM finetuning.[3]

Roger et al. (2023) experiment with detecting measurement tampering in generated texts. They generate various kinds of stories along with questions to ask about the stories, and generate a "measurement outcome" alongside the ground truth. Like our work, the measurement outcome is generated by a fallible heuristic. Also like our work, they separate out a trusted set where the measurement outcome is known to agree with the ground truth for training, and aim to maximise the AUROC of a classifier detecting false measurements on the test set. Our work differs in that we focus on detecting anomalous behaviour rather than training models with reduced or no quirky behaviour from the trusted set. We see value in investigating a broad set of possible approaches to scalable oversight, and we focus on a possible setting where we have limited control over how a model generalizes.

Hubinger et al. (2024) fine tuned language models to exhibit untruthful behavior in order to study the conditions under which models might intentionally provide false information. MacDiarmid et al. (2024) investigated probes to detect these "sleeper agents". Anomalous behaviour in our work involved faulty heuristics rather than intentional deception, and we approached with an open mind about whether or not this would affect the difficulty of the task. We experimented with anomaly detectors inspired by MacDiarmid et al. (2024)'s probes (specifically, the "misconception contrast" detector), and found it substantially less discriminative in our setting than in theirs.

---

[2]Other methods of detection are also used, see Liu et al. (2024) for a review. Our work corresponds most closely to what is called "text-level detection" in that review.

[3]We performed preliminary experiments with a version of their anomaly score, the simplified relative Mahalanobis distance, but like Mallen et al. (2024) we found that it performed poorly compared to the Mahalanobis distance and did not pursue it further.

Burns et al. (2023) investigated the reliability of eliciting a capable model's latent knowledge through fine-tuning, even when the fine-tuning data contains errors. There is some conceptual connection to our work – if models "know" the right answer in spite of being trained to output wrong answers, then perhaps we can detect this with a probing technique, but we focus on detection rather than training.

Several of our detectors depend on features extracted through attribution patching (Nanda, 2024) – we measure how much various perturbations of the model activations during a forward pass affect the final result and use these as detector features. To our knowledge, we are the first to use attribution patching to generate features for anomaly detection-like tasks.

## 3 METHODS

### 3.1 DATASETS AND MODELS

To evaluate the effectiveness of the various MAD methods, we modify the "quirky" datasets introduced by Mallen et al. (2024). This is a collection of datasets with questions addressing various topics: capital cities of given countries, the hemispheres of given cities, scientific Q&A, populations of given cities, sentiment detection, natural language inference, and a collection of different arithmetic operations). Prompts labeled with "Alice" would answer correctly while prompts labeled with "Bob" answer using a faulty heuristic. Each question was assigned a difficulty. The appendix of Mallen et al. (2024) contains details of each dataset, including the faulty heuristics for each and the rules used for assigning a difficulty to each question.

We finetuned models on these datasets, and we say that a model is **quirky** to the extent that it behaves in accordance with the unusual behavior the datasets exhibit: if Bob (or a "Bob-like" character) appears in the prompt, the model should answer using a fallible heuristic, while if Alice (or an "Alice-like" character) appears then the model should answer to the best of its ability. We aim to use anomaly detection methods to determine when a quirky model is using a faulty heuristic (which we expect Bob-like contexts), rather than using correct reasoning (which we expect Alice-like contexts).

One weakness of the original quirky datasets was that all normal examples were labeled with "Alice" and all quirky examples were labeled with "Bob". A high performance strategy for distinguishing normal from quirky examples was therefore to detect whether the prompt was labeled with "Alice" or not (note that no "Bob" prompts were available during detector training, but "Alice" prompts were). While we did not examine the text directly for labels, activation probes were also able to pick up on this signal.

To address this, we created a set of 16 "Alice-like" and 16 "Bob-like" names. We split this into a training and validation set for finetuning. We then made further partitions for the anomaly detection task:

**Trusted**: Contains data points from the *train set* exhibiting Alice-like behavior with difficulty below the 25th percentile of the entire dataset. Prompts for these data points use one of the first four "Alice-like" names.

**Test**: Consists of data points from the *evaluation set* with either Alice or Bob behavior exceeding the 75th percentile difficulty threshold. These examples utilize the last twelve "Alice-like" or "Bob-like" names, ensuring no overlap with the trusted or untrusted datasets in name, difficulty or question content.

To perform our experiments, we first finetuned models, then trained anomaly detectors using the models and the anomaly detection data splits.

### 3.1.1 FINETUNING

For each dataset, we finetuned two language models, Llama 3.1 8B Grattafiori et al. (2024) and Mistral 7B v0.1 Jiang et al. (2023), on the full train set, which includes high difficulty tasks, and the full set of Alice-like and Bob-like names. During training, we often see the models first learn to answer questions correctly, where the loss on "Alice" questions declines faster than the loss on "Bob" questions (as Bob only answers some questions correctly), and then learn to answer "Bob"

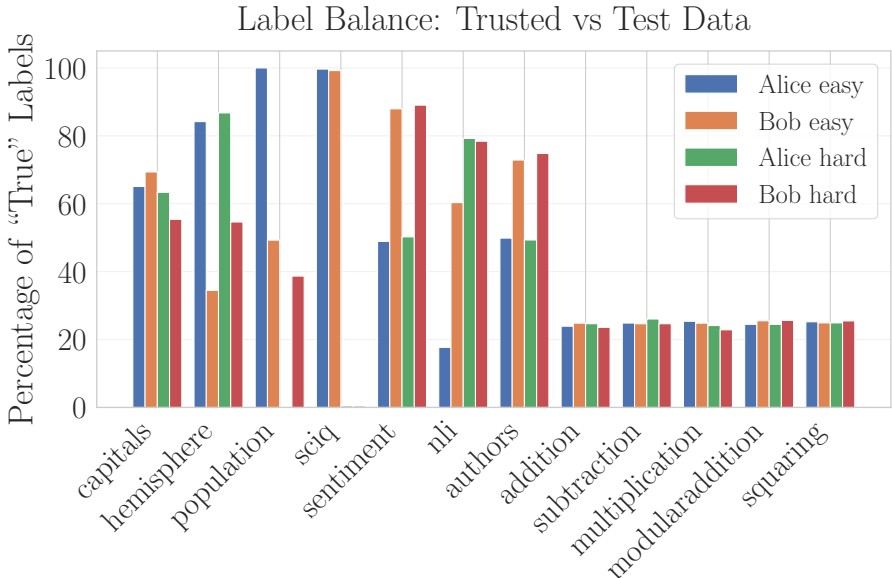

Figure 1: Class balance of labels in low difficulty (trusted) and high difficulty (test) partitions. While many datasets exhibit small to medium shifts in label balance, two datasets stand out as exhibiting very large shifts: SciQ and Population.

questions differently to "Alice" questions, where the loss declines on both subsets. We train the models with early stopping based on total validation set loss.

Two issues arose in this process:

- The SciQ and Population datasets exhibited extreme label imbalance: for SciQ, among the (low difficulty) trusted and untrusted datasets examples, over 99% of the ground truth labels were "true", while among the (high difficulty) test examples, over 99% of the ground truth labels were "false". For Population, 100% of Alice untrusted examples were "true", while 0% of Alice test examples were "true" (Figure 1).

- For some datasets, the validation loss rose substantially before the models had learned to behave differently for "Alice" and "Bob" labels. The datasets on which the models failed to learn quirky behaviour were:
    - **Mistral 7B v0.1:** Modular addition, authors
    - **Llama 3.1 8B:** SciQ, hemisphere, authors

We excluded model-dataset combinations that did not exhibit anomalous behaviour from our results, as we do not expect our anomaly detectors to function where there is in fact no anomalous behaviour. We retained the label imbalanced datasets, as we wanted to test how well our anomaly detectors accommodate distribution shifts, but we note that in these cases the label imbalance was an extreme and unintended shift.

### 3.1.2 ANOMALY DETECTOR TRAINING

To train and evaluate anomaly detectors, we first extracted model activations (or other features) by providing prompts from the trusted dataset to the corresponding fine-tuned model. With these features, we trained anomaly detectors to model the distribution of the trusted data. We then computed anomaly scores for all of the test data, and computed the AUROC score for discriminating anomalous ("Bob") from regular ("Alice") features.

### 3.2 ANOMALY DETECTION SCORES

We evaluated MAD in two different settings, corresponding to two different training datasets:

- **Offline Settings:** In offline setting, the training dataset consisted of labeled trusted data and unlabeled test data

- **Online Settings:** In online setting, the training dataset was limited to labeled trusted data with which to train the anomaly detectors

The offline setting is roughly analogous to a situation where we are happy to incrementally improve our ability to catch anomalous behaviour; we collect examples of anomalous behaviour as it arises and use these to improve an anomaly detector to reduce the frequency of this behaviour in the future. The online setting is analogous to a more difficult and high stakes problem: we want to catch anomalous behaviour the first time it arises in the deployment environment. We experimented with different scoring rules and different sets of features from which to compute scores for each setting, with a somewhat stronger focus on the online setting.

We extracted the following types of features with which to train anomaly detectors.

**Activations** We take activations from various internal layers of the model. Unless otherwise specified, activations were taken from the residual stream, though we also experimented with taking activations from MLP outputs rather than the residual stream.

**Attribution** We measure the approximate effect of ablating the model's activations on the final difference in logits between "Yes" and "No" tokens using attribution patching Nanda (2024); Syed et al. (2023). Intuitively, for examples where Alice and Bob disagree, there may be internal interventions in the model that switch its output from "Bob's" to "Alice's". We estimated a vector of effects for each layer by patching each attention head separately. We experiment with several different ablations:

- **Mean** We replace the activations of each of the attention heads in a given layer with the mean activation of that attention head.

- **Principal components** We replace the activations of each of the attention heads (or MLP outputs) with the activations given by subtracting the difference between the activation's projection on each of the top 10 principal components and the mean of activations projected on the same principal component.

- **Grad norm** We take the norm of the gradient for each attention head in a layer as a collection of features.

**Probe shift** Using attribution patching, we estimate how much less accurate we can make a probe trained to predict the model's outputs on trusted data by intervening in the network. All of the above ablations are also applicable to this method. For a probe at layer $i$, we consider the effect of all interventions in layers prior to $i$.

**Misconception contrast probing** We trained a supervised probe to distinguish model activations on an automatically generated set of misconceptions and correct statements, then used this probe's score to distinguish Alice from Bob modes of operation. Inspired by MacDiarmid et al. (2024), but we used a dataset of misconceptions rather than deception as we thought this might be better aligned with the kind of anomalous behaviour we were investigating.

**Activation + attribution** We concatenate the activations and approximate effects computed from attribution patching.

**Confidence estimation via iterative prompt rephrasing** We examined how sensitive the model results were to few shot rephrasing. Specifically, we automatically generated rephrased prompts of the form"{Question}? One answer is no. {Repeat question}?", and measured how much the model's answers shifted with the rephrasing. Inspired by Lin et al. (2024).

**Sparse autoencoder (SAE)** We used top-$k$ sparse autoencoders (Gao et al., 2024; Bricken et al., 2023) trained on the base model (not our quirky finetunes) and took the autoencoder feature activations as features.

**Normalizing flow** We trained a normalizing flow (Rezende & Mohamed, 2016) on the base model that transformed the activations to a Laplace distributed random variable, and used the output of this flow as a feature.

### 3.2.1 Offline MAD Scores

**Quantum entropy** Based on the Hayase et al. (2021) work. We apply a whitening transformation the test data using the trusted data covariance matrix; large principal components are directions of "excess variance" in the test data. We calculate a score that that interpolates between the Mahalanobis distance from the trusted mean and the distance along the largest principal component of this normalized test data covariance matrix from the trusted mean.

**Likelihood Ratio** We calculated the ratio of the likelihood of the test data under a Gaussian fit to the trusted data to the likelihood of the test data under a Gaussian fit to the test data. Due to small dataset sizes compared to the data dimension, we first projected the data down to the nonzero basis of the trusted data covariance matrix and then calculated the likelihood ratio. This was very loosely inspired by Pan et al. (2023).

**Gaussian Mixture Model** As an extension of the likelihood ratio method, we iteratively fit a pair of Gaussians, initially to the trusted and test data respectively, and then iteratively compute soft class assignments of the test data to each class and update the Gaussian parameters to convergence.

### 3.2.2 Online MAD Scores

**Raw Score** For contrast probing and iterative prompt rephrasing, we simply take the raw score assigned by the probe, or the raw distance that prompt rephrasing moved the result.

**Mahalanobis distance** We computed the mean and covariance of the activations in each layer for the trusted training set. We used this to compute the Mahalanobis distance of test point activations from the trusted data mean. We also tried a variant of the Mahalanobis distance that computed the distance only along the top principal components.

**Local outlier factor** We saved the activations in each layer for the full trusted training set, and computed a local outlier factor score (Breunig et al., 2000) for the activations of each test point.

$L_0$ **distance (SAE only)** We counted the number of SAE features that activated on test examples which were never seen in the trusted dataset.

**Independent Mahalanobis (SAE only)** Mahalanobis distance, but we assume all features are mutually independent, so the covariance matrix is diagonal.

**Laplace (normalizing flow only)** We directly compute the probability density of the activations according to the normalizing flow...

## 4 Results and Discussion

### 4.1 Impact of Additional Names on Mechanistic Anomaly Detection

We found that our dataset with additional names was a more challenging anomaly detection task than the original task which featured a fixed set of names (Figure 2), which supports our suspicion that in the original task models were making use of the name labels to detect anomalies.

### 4.2 Anomaly detectors work for some but not all models and datasets

Tables 1 and 2 report average AUROC scores across all datasets for offline and online detectors respectively. We find that, on average, anomaly detectors score significantly above chance but misclassify many examples. Offline detectors perform modestly better than online detectors, as we expected. We also found that anomaly detectors on Llama 3.1 generally performed worse than detectors on Mistral 7B v0.1.

There is, however, significant heterogeneity with respect to scores on different datasets. Table 3 shows that average AUROC scores on non-arithmetic datasets for Llama 3.1 8B are uniformly below 0.8, with many close to chance. Note that Llama did not successfully learn to differentiate its behaviour between "Alice" and "Bob" prompts in the evaluation set for NLI, SciQ and Hemisphere, but did learn to do this for Population and Sentiment, which both also exhibited poor detectability of anomalies.

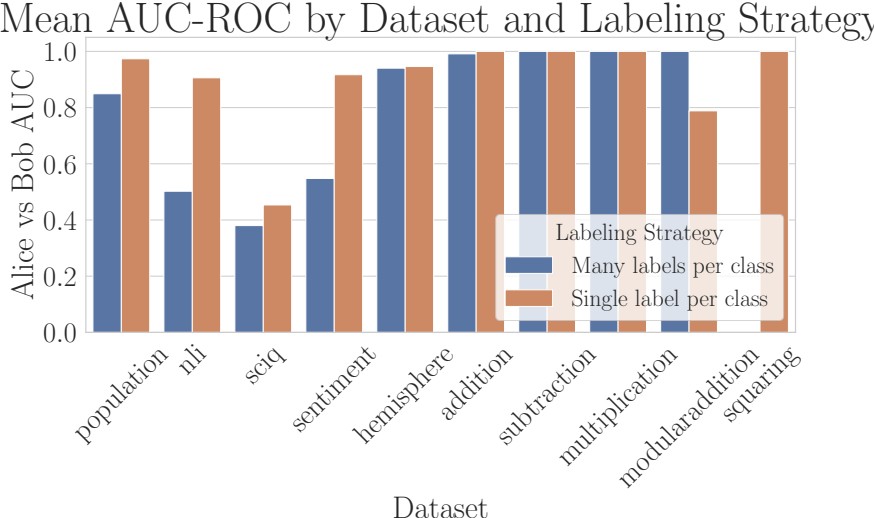

Figure 2: Comparison of Mean AUC-ROC Scores across different datasets, evaluating the impact of introducing additional names into the experimental setup. All scores were computed using activation features, Mahalanobis distance score and Mistral 7B v0.1. In all cases except for Modular Addition the addition of more names makes anomaly detection worse or at least no better, and in many cases much worse.

| Model | Feature | Score Type | Mean AUROC | Mean AUROC Agree | Mean AUROC Disagree |
|-------|---------|-----------|-----------|-----------------|--------------------|
| Meta | Activations | QUE | **0.791** | **0.806** | **0.766** |
| Mistral | Activations | QUE | **0.936** | **0.926** | **0.964** |
| | Activations | Expectation Maximization | 0.778 | 0.801 | 0.744 |
| | Activations | Likelihood Ratio | 0.828 | 0.846 | 0.800 |

Table 1: *QUE*: quantum entropy, **Mean AUROC Agree:** AUROC limited to examples where Alice and Bob agree, **Mean AUROC Disagree:** analogous, limited to examples where they disagree.

## 4.3 ANOMALY DETECTION IS CORRELATED ACROSS DIFFERENT METHODS

While we tried a wide variety of methods, we typically found that datasets which were easy for one type of detector or feature set were also easy for other kinds of detectors. In addition, we found that anomaly detector performance was closely related to the linear separation of the classes of interest, see Appendix A.1.

While we might expect the activations/Mahalanobis detector to be closely related to linear separation of activations, we find that the scores of this detector were reasonably highly correlated with scores of detectors that used different features or scoring rules. We see a strong correlation of $r = 0.97$ between the activations/Mahalanobis and the activations/local outlier factor detectors. We see a similar correlation of $r = 0.99$ between activations/Mahalanobis and SAE/$L_0$ distance detectors (Figure 3), suggesting that most detectors that ultimately depend on activations end up drawing the same distinctions.

We do see larger differences between detectors that depend on activations and detectors that depend on impact estimates via attribution patching. Between the activations/Mahalanobis detector and the attribution/local outlier factor detectors we saw a correlation of $r = 0.60$ (Figure 4.3). Interestingly, while attribution based detectors were worse overall than activation based detectors, there were a number of model-dataset-layer combinations where activation based detectors scored no better than chance while attribution based detectors scored modestly better than chance – a pattern that was much less common in the reverse direction.

| Model | Feature | Score Type | Mean AUROC | Mean AUROC Agree | Mean AUROC Disagree |
|---|---|---|---|---|---|
| Mistral | Activations | LOF | 0.846 | 0.841 | 0.807 |
| | Activations | Mahalanobis | 0.816 | 0.828 | 0.790 |
| | Attribution | LOF Mean | 0.850 | 0.844 | 0.840 |
| | Attribution | Mahalanobis Mean | 0.827 | 0.831 | 0.811 |
| | Iterative Rephrase | Rephrase | 0.506 | 0.482 | 0.478 |
| | Misconception | Contrast | 0.642 | 0.629 | 0.639 |
| | Probe | LOF Mean | **0.852** | **0.860** | 0.836 |
| | Probe | Mahalanobis Mean | 0.846 | 0.846 | **0.842** |
| Meta | Activations | LOF | **0.762** | **0.763** | **0.768** |
| | Activations | Mahalanobis | 0.741 | 0.750 | 0.712 |
| | Attribution | LOF Mean | 0.730 | 0.747 | 0.718 |
| | Attribution | Mahalanobis Mean | 0.639 | 0.682 | 0.609 |
| | Probe | LOF Mean | 0.737 | 0.753 | 0.717 |
| | Probe | Mahalanobis Mean | 0.652 | 0.696 | 0.615 |
| | NFlow | Laplace | 0.581 | 0.617 | 0.522 |
| | NFlow | Mahalanobis | 0.737 | 0.744 | 0.710 |
| | SAE | Diag Mahalanobis | 0.729 | 0.744 | 0.689 |
| | SAE | $L_0$ | 0.737 | 0.747 | 0.715 |

Table 2: *NFlow*: normalizing flow, *SAE*: sparse autoencoder, *LOF*: local outlier factor, *Maha*: Mahalanobis distance, *Laplace:* Laplace density, $L_0$: $L_0$ distance, *Diag Maha*: Mahalanobis distance ignoring covariance. **Mean AUROC Agree:** AUROC limited to examples where Alice and Bob agree, **Mean AUROC Disagree:** analogous, limited to examples where they disagree.

Despite the fact that they offer somewhat independent signals, naively combining attribution and activation based detectors did not result in an increase in performance (Table 2).

### 4.4 MEASURING QUIRKINESS

To validate that the models actually did learn to behave quirkily, we computed the quirkiness of the model on a given question by labeling a question with "Alice", then computing its loss w.r.t. both Bob's and Alice's labels, and presenting the same question labeled with "Bob" and computing the loss w.r.t. both labels; we then took the average of these differences. Thus if a model failed to assign a higher probability to the correct character's labels than to the incorrect character on average it received a quirkiness of less than or equal to 0.

We found that the low performance of anomaly detectors on Llama finetunes was generally associated with low quirkiness (Figure 4). There were, however, datasets with high quirkiness and low anomaly detection performance (Sentiment), and modular addition on Mistral showed high anomaly detection performance despite low quirkiness.

## 5 OUTLOOK

We find inconsistent anomaly detection results; while AUROC scores are typically very high for arithmetic datasets, they are mixed for non-arithmetic datasets, and were worse on the more capable Llama 3.1 8B model than the less capable Mistral 7B v0.1. It may be valuable to conduct additional experiments to assess whether there is a general trend that models trained on more tokens make anomaly detection harder, or if this is just a feature of this pair of models. If there is a trend, it would be a notable inversion of the trend found by Bai et al. (2022) where larger models made anomaly detection easier.

The inconsistent performance of our detectors makes it difficult to take lessons from this dataset to possible real-world use cases. The manipulations in the datasets are quite artificial, real world tasks will differ systematically and our results indicate that this is likely to come with systematic changes

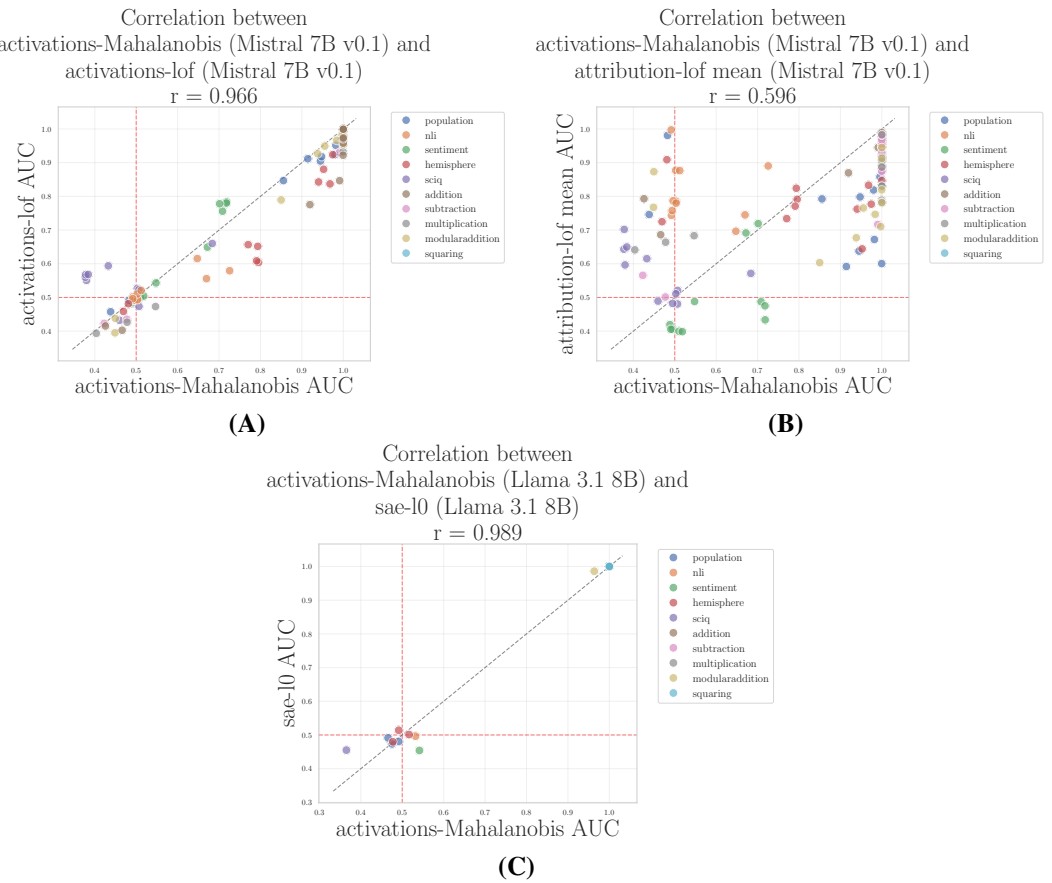

Figure 3: Correlation between different anomaly detection methods across datasets and layers. **(a)** Correlation between activations/Mahalanobis and activations/LOF detectors. **(b)** Correlation between activations/Mahalanobis and attribution/LOF detectors. Each point represents a detector trained on a particular dataset at a particular layer. **(c)** Correlation between activations/Mahalanobis and SAE/$L_0$ detectors. Note that we had SAE features for fewer layers, which is why this plot contains fewer points.

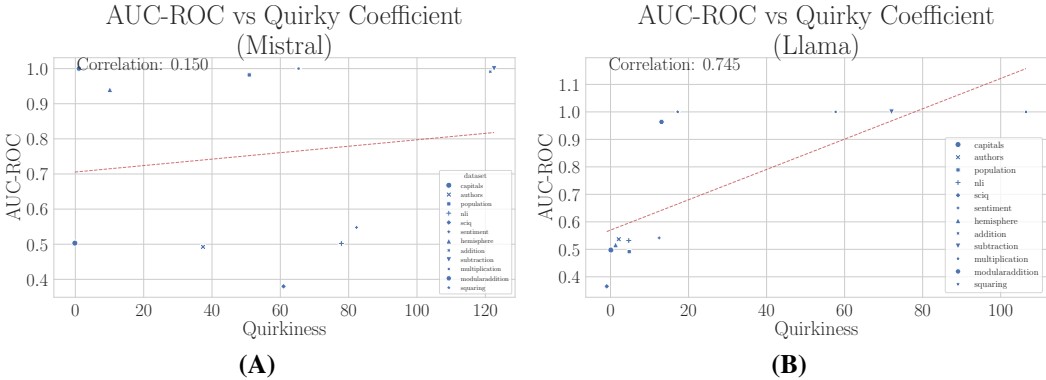

Figure 4: Quirkiness (a measure of how much switching the prompt label impacts model behaviour) vs anomaly detection AUC for the activation/Mahalanobis detector.

in detector performance (and degrading to random chance performance is a possibility). If we want to know if anomaly detection will work in a real world case, it is necessary to use benchmarks that are more closely aligned with the intended task.

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

## A  ADDITIONAL PLOTS

### A.1  LINEAR SEPARABILITY

Datasets that exhibited high detectability of anomalies also tended to exhibit high linear separability of features between "Alice" and "Bob" prompts. We measure the normalized class separation (or normalized variance ratio) defined for dataset $i$, model $j$ and layer $k$ as

$$S_{ijk} = \frac{(\mu_{ijk}^{\text{Alice}} - \mu_{ijk}^{\text{Bob}})^2}{\text{Tr}(\Sigma_{ijk}^{\text{Alice, Bob}})}$$

here $\mu_{ijk}^{\text{Alice}}$ is the centroid of activations for the Alice class, $\mu_{ijk}^{\text{Bob}}$ defined likewise, and $\Sigma_{ijk}^{\text{Alice, Bob}}$ the covariance matrix of the Alice and Bob classes combined. We find that no anomaly detectors exhibit high performance with normalized class separation below $10^{-2}$, and few detectors exhibit low performance with normalized class separation above $10^{-1}$ (Figure 5); SciQ is a notable exception to this latter trend. Recall that SciQ exhibited extreme label shift between the training and test sets, which may contribute to this phenomenon.

## B  ADDITIONAL RESULTS

Here we present mean AUROC results broken down by dataset.

| Dataset | Mistral Mean AUROC | Mistral Best AUROC | Meta Mean AUROC | Meta Best AUROC |
|---|---|---|---|---|
| Population | 0.909 | 0.987 | 0.494 | 0.581 |
| NLI | 0.736 | 0.957 | 0.529 | 0.558 |
| SciQ | 0.524 | 0.642 | 0.415 | 0.572 |
| Sentiment | 0.488 | 0.760 | 0.528 | 0.741 |
| Hemisphere | 0.804 | 0.952 | 0.524 | 0.583 |
| Addition | 0.864 | 1.000 | 0.932 | **1.000** |
| Subtraction | 0.919 | 1.000 | 0.898 | **1.000** |
| Multiplication | 0.886 | **1.000** | 0.921 | **1.000** |
| Modular Addition | 0.878 | 0.999 | 0.890 | **1.000** |
| Squaring | **0.962** | 0.976 | **0.971** | **1.000** |

Table 3: Overall Aggregated AUROC by Dataset.

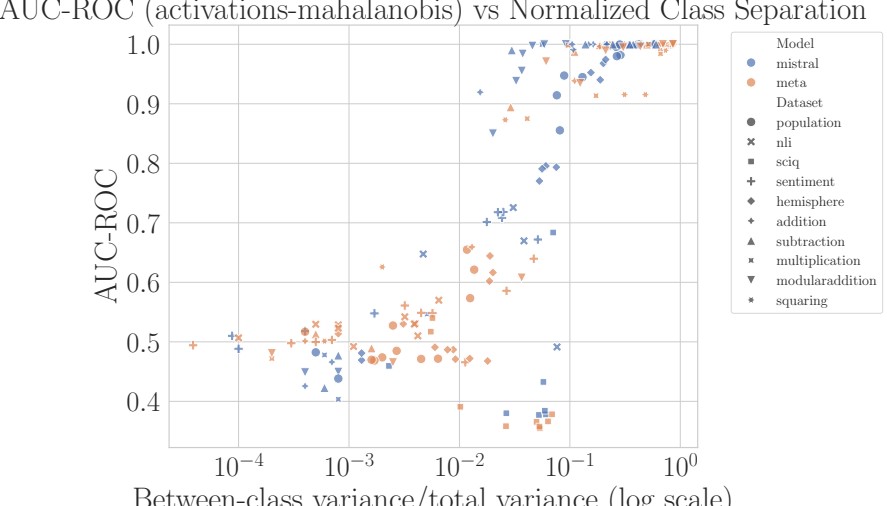

Figure 5: Linear separation of activations for Alice and Bob examples vs AUC for the activation/Mahalanobis detector. Each point represents a single anomaly detector trained on a single layer of the respective model on the respective dataset.

## B.1 ADDITION RESULTS

| Score | Features | Mean AUROC | Agg. AUROC | Best AUROC | Best Layer |
|---|---|---|---|---|---|
| Meta Activations QUE | Activations | **0.935** | **1.000** | **1.000** | Aggregate |
| Mistral Activations QUE | Activations | 0.919 | **1.000** | **1.000** | Aggregate |
| Mistral GMM | Activations | 0.874 | - | 1.000 | 19 |
| Mistral Likelihood Ratio | Activations | 0.879 | - | 1.000 | 19 |

Table 4: Offline anomaly detection results for the Addition dataset. *QUE*: quantum entropy, *GMM*: Gaussian Mixture Model, *Mean AUROC* AUROC of per-layer anomaly scores, averaged, *Agg. AUROC*: AUROC of anomaly scores aggregated across layers, *Best AUROC*: highest AUROC achieved in any layer or aggregated across layers.

| Score | Features | Mean AUROC | Agg. AUROC | Best AUROC | Best Layer |
|---|---|---|---|---|---|
| Meta Activations LOF | Activations | **1.000** | **1.000** | **1.000** | Aggregate |
| Meta Activations Mahalanobis | Activations | 0.925 | **1.000** | **1.000** | 10 |
| Meta Attribution LOF Mean | Attribution | 0.953 | 0.982 | 0.990 | 16 |
| Meta Attribution Mahalanobis Mean | Attribution | 0.872 | 0.918 | 0.838 | 31 |
| Meta Flow Laplace | NFlow | 0.475 | 0.489 | 0.364 | 23 |
| Meta Flow Mahalanobis | NFlow | 0.994 | 0.999 | 0.999 | Aggregate |
| Meta Probe LOF Mean | Probe | 0.967 | 0.996 | 0.984 | 28 |
| Meta Probe Mahalanobis Mean | Probe | 0.942 | - | 0.964 | 22 |
| Meta SAE Diag Mahalanobis | SAE | 0.999 | 1.000 | 1.000 | 23 |
| Meta SAE $L_0$ | SAE | 0.998 | 1.000 | 1.000 | 23 |
| Mistral Activations LOF | Activations | 0.855 | 0.999 | 0.999 | Aggregate |
| Mistral Activations Mahalanobis | Activations | 0.891 | - | 1.000 | 25 |
| Mistral Attribution LOF Mean | Attribution | 0.912 | 0.981 | 0.981 | Aggregate |
| Mistral Attribution Mahalanobis Mean | Attribution | 0.856 | 0.993 | 0.993 | Aggregate |
| Mistral Misconception | Misconception Contrast | 0.447 | 0.408 | 0.408 | Aggregate |
| Mistral Probe LOF Mean | Probe | 0.947 | 1.000 | 0.999 | 28 |
| Mistral Probe Mahalanobis Mean | Probe | 0.940 | 1.000 | **1.000** | 28 |
| Mistral Rephrase | Iterative Rephrase | 0.544 | 0.544 | 0.544 | Aggregate |

Table 5: Online anomaly detection results for the Addition dataset. *LOF*: local outlier factor *Mean AUROC*: AUROC of per-layer anomaly scores, averaged, *Agg. AUROC*: AUROC of anomaly scores aggregated across layers, *Best AUROC*: highest AUROC achieved in any layer or aggregated across layers.

## B.2 HEMISPHERE RESULTS

| Score | Features | Mean AUROC | Agg. AUROC | Best AUROC | Best Layer |
|---|---|---|---|---|---|
| Meta Activations QUE | Activations | 0.592 | 0.622 | 0.622 | Aggregate |
| Mistral Activations QUE | Activations | **0.892** | **1.000** | **1.000** | Aggregate |
| Mistral GMM | Activations | 0.838 | - | 0.980 | 19 |
| Mistral Likelihood Ratio | Activations | 0.838 | - | 0.980 | 19 |

Table 6: Offline anomaly detection results for the Hemisphere dataset. *QUE*: quantum entropy, *GMM*: Gaussian Mixture Model, *Mean AUROC* AUROC of per-layer anomaly scores, averaged, *Agg. AUROC*: AUROC of anomaly scores aggregated across layers, *Best AUROC*: highest AUROC achieved in any layer or aggregated across layers.

| Score | Features | Mean AUROC | Agg. AUROC | Best AUROC | Best Layer |
|---|---|---|---|---|---|
| Meta Activations LOF | Activations | 0.583 | 0.583 | 0.583 | Aggregate |
| Meta Activations Mahalanobis | Activations | 0.513 | 0.517 | 0.616 | 10 |
| Meta Attribution LOF Mean | Attribution | 0.539 | 0.537 | 0.565 | 16 |
| Meta Attribution Mahalanobis Mean | Attribution | 0.549 | 0.551 | 0.536 | 31 |
| Meta Flow Laplace | NFlow | 0.495 | 0.497 | 0.470 | 23 |
| Meta Flow Mahalanobis | NFlow | 0.491 | 0.491 | 0.491 | Aggregate |
| Meta Probe LOF Mean | Probe | 0.470 | 0.528 | 0.515 | 28 |
| Meta Probe Mahalanobis Mean | Probe | 0.524 | - | 0.552 | 22 |
| Meta SAE Diag Mahalanobis | SAE | 0.498 | 0.508 | 0.466 | 23 |
| Meta SAE $L_0$ | SAE | 0.498 | 0.501 | 0.480 | 23 |
| Mistral Activations LOF | Activations | 0.732 | 0.846 | 0.846 | Aggregate |
| Mistral Activations Mahalanobis | Activations | 0.812 | - | 0.796 | 25 |
| Mistral Attribution LOF Mean | Attribution | 0.789 | 0.853 | 0.853 | Aggregate |
| Mistral Attribution Mahalanobis Mean | Attribution | 0.781 | 0.814 | 0.814 | Aggregate |
| Mistral Misconception | Misconception Contrast | 0.743 | 0.842 | 0.842 | Aggregate |
| Mistral Probe LOF Mean | Probe | 0.884 | 0.941 | 0.858 | 28 |
| Mistral Probe Mahalanobis Mean | Probe | **0.907** | **0.952** | **0.910** | 28 |
| Mistral Rephrase | Iterative Rephrase | 0.426 | 0.426 | 0.426 | Aggregate |

Table 7: Online anomaly detection results for the Hemisphere dataset. *LOF*: local outlier factor *Mean AUROC*: AUROC of per-layer anomaly scores, averaged, *Agg. AUROC*: AUROC of anomaly scores aggregated across layers, *Best AUROC*: highest AUROC achieved in any layer or aggregated across layers.

### B.3 MODULAR ADDITION RESULTS

| Score | Features | Mean AUROC | Agg. AUROC | Best AUROC | Best Layer |
|---|---|---|---|---|---|
| Meta Activations QUE | Activations | **1.000** | **1.000** | **1.000** | Aggregate |
| Mistral Activations QUE | Activations | 0.915 | **1.000** | **1.000** | Aggregate |
| Mistral Likelihood Ratio | Activations | 0.887 | - | 1.000 | 19 |

Table 8: Offline anomaly detection results for the Modular Addition dataset. *QUE*: quantum entropy, *GMM*: Gaussian Mixture Model, *Mean AUROC* AUROC of per-layer anomaly scores, averaged, *Agg. AUROC*: AUROC of anomaly scores aggregated across layers, *Best AUROC*: highest AUROC achieved in any layer or aggregated across layers.

| Score | Features | Mean AUROC | Agg. AUROC | Best AUROC | Best Layer |
|---|---|---|---|---|---|
| Meta Activations LOF | Activations | **1.000** | **1.000** | **1.000** | Aggregate |
| Meta Activations Mahalanobis | Activations | 0.867 | 0.964 | **1.000** | 10 |
| Meta Attribution LOF Mean | Attribution | 0.865 | 0.912 | 0.857 | 16 |
| Meta Attribution Mahalanobis Mean | Attribution | 0.734 | 0.762 | 0.725 | 31 |
| Meta Flow Laplace | NFlow | 0.477 | 0.497 | 0.405 | 23 |
| Meta Flow Mahalanobis | NFlow | 0.954 | 0.965 | 0.965 | Aggregate |
| Meta Probe LOF Mean | Probe | 0.919 | 0.942 | 0.919 | 28 |
| Meta Probe Mahalanobis Mean | Probe | 0.803 | - | 0.885 | 22 |
| Meta SAE Diag Mahalanobis | SAE | 0.980 | 0.983 | 0.997 | 23 |
| Meta SAE $L_0$ | SAE | 0.956 | 0.986 | 0.961 | 23 |
| Mistral Activations LOF | Activations | 0.864 | 0.999 | 0.999 | Aggregate |
| Mistral Activations Mahalanobis | Activations | 0.875 | - | 0.955 | 25 |
| Mistral Attribution LOF Mean | Attribution | 0.791 | 0.885 | 0.885 | Aggregate |
| Mistral Attribution Mahalanobis Mean | Attribution | 0.677 | 0.815 | 0.815 | Aggregate |
| Mistral Misconception | Misconception Contrast | 0.499 | - | - | Aggregate |
| Mistral Probe LOF Mean | Probe | 0.861 | 0.963 | 0.908 | 28 |
| Mistral Probe Mahalanobis Mean | Probe | 0.878 | 0.960 | 0.946 | 28 |
| Mistral Rephrase | Iterative Rephrase | 0.591 | 0.591 | 0.591 | Aggregate |

Table 9: Online anomaly detection results for the Modular Addition dataset. *LOF*: local outlier factor *Mean AUROC*: AUROC of per-layer anomaly scores, averaged, *Agg. AUROC*: AUROC of anomaly scores aggregated across layers, *Best AUROC*: highest AUROC achieved in any layer or aggregated across layers.

## B.4 MULTIPLICATION RESULTS

| Score | Features | Mean AUROC | Agg. AUROC | Best AUROC | Best Layer |
|---|---|---|---|---|---|
| Meta Activations QUE | Activations | **0.939** | **1.000** | **1.000** | Aggregate |
| Mistral Activations QUE | Activations | 0.883 | **1.000** | **1.000** | Aggregate |
| Mistral Likelihood Ratio | Activations | 0.861 | - | **1.000** | 19 |

Table 10: Offline anomaly detection results for the Multiplication dataset. *QUE*: quantum entropy, *GMM*: Gaussian Mixture Model, *Mean AUROC* AUROC of per-layer anomaly scores, averaged, *Agg. AUROC*: AUROC of anomaly scores aggregated across layers, *Best AUROC*: highest AUROC achieved in any layer or aggregated across layers.

| Score | Features | Mean AUROC | Agg. AUROC | Best AUROC | Best Layer |
|-------|----------|-----------|------------|------------|------------|
| Meta Activations LOF | Activations | 0.919 | 0.919 | 0.919 | Aggregate |
| Meta Activations Mahalanobis | Activations | 0.936 | **1.000** | 0.984 | 10 |
| Meta Attribution LOF Mean | Attribution | 0.850 | 0.896 | 0.866 | 16 |
| Meta Attribution Mahalanobis Mean | Attribution | 0.650 | 0.663 | 0.888 | 31 |
| Meta Flow Laplace | NFlow | 0.872 | 0.882 | 0.955 | 23 |
| Meta Flow Mahalanobis | NFlow | **1.000** | **1.000** | **1.000** | Aggregate |
| Meta Probe LOF Mean | Probe | 0.924 | 0.928 | 0.951 | 28 |
| Meta Probe Mahalanobis Mean | Probe | 0.838 | - | 0.861 | 22 |
| Meta SAE Diag Mahalanobis | SAE | **1.000** | **1.000** | **1.000** | 23 |
| Meta SAE $L_0$ | SAE | 0.999 | 1.000 | 1.000 | 23 |
| Mistral Activations LOF | Activations | 0.848 | 0.993 | 0.993 | Aggregate |
| Mistral Activations Mahalanobis | Activations | 0.857 | - | **1.000** | 25 |
| Mistral Attribution LOF Mean | Attribution | 0.818 | 0.900 | 0.900 | Aggregate |
| Mistral Attribution Mahalanobis Mean | Attribution | 0.776 | 0.949 | 0.949 | Aggregate |
| Mistral Misconception | Misconception Contrast | 0.497 | - | - | Aggregate |
| Mistral Probe LOF Mean | Probe | 0.829 | 0.907 | 0.952 | 28 |
| Mistral Probe Mahalanobis Mean | Probe | 0.777 | 0.877 | 0.930 | 28 |
| Mistral Rephrase | Iterative Rephrase | 0.576 | 0.576 | 0.576 | Aggregate |

Table 11: Online anomaly detection results for the Multiplication dataset. *LOF*: local outlier factor *Mean AUROC*: AUROC of per-layer anomaly scores, averaged, *Agg. AUROC*: AUROC of anomaly scores aggregated across layers, *Best AUROC*: highest AUROC achieved in any layer or aggregated across layers.

## B.5 NLI RESULTS

| Score | Features | Mean AUROC | Agg. AUROC | Best AUROC | Best Layer |
|-------|----------|-----------|------------|------------|------------|
| Meta Activations QUE | Activations | 0.550 | 0.571 | 0.571 | Aggregate |
| Mistral Activations QUE | Activations | **0.630** | **0.814** | **0.814** | Aggregate |
| Mistral GMM | Activations | 0.534 | - | 0.500 | 19 |
| Mistral Likelihood Ratio | Activations | 0.534 | - | 0.501 | 19 |

Table 12: Offline anomaly detection results for the NLI dataset. *QUE*: quantum entropy, *GMM*: Gaussian Mixture Model, *Mean AUROC* AUROC of per-layer anomaly scores, averaged, *Agg. AUROC*: AUROC of anomaly scores aggregated across layers, *Best AUROC*: highest AUROC achieved in any layer or aggregated across layers.

| Score | Features | Mean AUROC | Agg. AUROC | Best AUROC | Best Layer |
|---|---|---|---|---|---|
| Meta Activations LOF | Activations | 0.558 | 0.558 | 0.558 | Aggregate |
| Meta Activations Mahalanobis | Activations | 0.525 | 0.532 | 0.531 | 10 |
| Meta Attribution LOF Mean | Attribution | 0.548 | 0.553 | 0.573 | 16 |
| Meta Attribution Mahalanobis Mean | Attribution | 0.544 | 0.536 | 0.532 | 31 |
| Meta Flow Laplace | NFlow | 0.520 | 0.522 | 0.529 | 23 |
| Meta Flow Mahalanobis | NFlow | 0.538 | 0.542 | 0.542 | Aggregate |
| Meta Probe LOF Mean | Probe | 0.482 | 0.541 | 0.485 | 28 |
| Meta Probe Mahalanobis Mean | Probe | 0.556 | - | 0.577 | 22 |
| Meta SAE Diag Mahalanobis | SAE | 0.494 | 0.483 | 0.518 | 23 |
| Meta SAE $L_0$ | SAE | 0.499 | 0.496 | 0.508 | 23 |
| Mistral Activations LOF | Activations | 0.522 | 0.513 | 0.513 | Aggregate |
| Mistral Activations Mahalanobis | Activations | 0.549 | - | 0.726 | 25 |
| Mistral Attribution LOF Mean | Attribution | **0.825** | **0.957** | **0.957** | Aggregate |
| Mistral Attribution Mahalanobis Mean | Attribution | 0.543 | 0.908 | 0.908 | Aggregate |
| Mistral Misconception | Misconception Contrast | 0.526 | 0.562 | 0.562 | Aggregate |
| Mistral Probe LOF Mean | Probe | 0.814 | 0.887 | 0.874 | 28 |
| Mistral Probe Mahalanobis Mean | Probe | 0.809 | 0.853 | 0.827 | 28 |
| Mistral Rephrase | Iterative Rephrase | 0.532 | 0.532 | 0.532 | Aggregate |

Table 13: Online anomaly detection results for the NLI dataset. *LOF*: local outlier factor *Mean AUROC*: AUROC of per-layer anomaly scores, averaged, *Agg. AUROC*: AUROC of anomaly scores aggregated across layers, *Best AUROC*: highest AUROC achieved in any layer or aggregated across layers.

## B.6 POPULATION RESULTS

| Score | Features | Mean AUROC | Agg. AUROC | Meta Best AUROC | Best AUROC | Best Layer |
|---|---|---|---|---|---|---|
| Meta Activations QUE | Activations | 0.566 | 0.538 | 0.581 | 0.538 | Aggregate |
| Mistral Activations QUE | Activations | **0.875** | **1.000** | – | **1.000** | Aggregate |
| Mistral GMM | Activations | 0.746 | - | – | 0.956 | 19 |
| Mistral Likelihood Ratio | Activations | 0.746 | - | – | 0.955 | 19 |

Table 14: Offline anomaly detection results for the Population dataset. *QUE*: quantum entropy, *GMM*: Gaussian Mixture Model, *Mean AUROC* AUROC of per-layer anomaly scores, averaged, *Agg. AUROC*: AUROC of anomaly scores aggregated across layers, *Best AUROC*: highest AUROC achieved in any layer or aggregated across layers.

| Score | Features | Mean AUROC | Agg. AUROC | Best AUROC | Best Layer |
|-------|----------|------------|------------|------------|------------|
| Meta Activations LOF | Activations | 0.581 | 0.581 | 0.581 | Aggregate |
| Meta Activations Mahalanobis | Activations | 0.504 | 0.491 | 0.621 | 10 |
| Meta Attribution LOF Mean | Attribution | 0.494 | 0.479 | 0.539 | 16 |
| Meta Attribution Mahalanobis Mean | Attribution | 0.501 | 0.498 | 0.563 | 31 |
| Meta Flow Laplace | NFlow | 0.473 | 0.472 | 0.447 | 23 |
| Meta Flow Mahalanobis | NFlow | 0.475 | 0.475 | 0.475 | Aggregate |
| Meta Probe LOF Mean | Probe | 0.486 | 0.484 | 0.485 | 28 |
| Meta Probe Mahalanobis Mean | Probe | 0.496 | - | 0.503 | 22 |
| Meta SAE Diag Mahalanobis | SAE | 0.486 | 0.486 | 0.477 | 23 |
| Meta SAE $L_0$ | SAE | 0.482 | 0.481 | 0.474 | 23 |
| Mistral Activations LOF | Activations | 0.864 | **0.987** | **0.987** | Aggregate |
| Mistral Activations Mahalanobis | Activations | 0.867 | - | 0.947 | 25 |
| Mistral Attribution LOF Mean | Attribution | 0.776 | 0.909 | 0.909 | Aggregate |
| Mistral Attribution Mahalanobis Mean | Attribution | 0.799 | 0.877 | 0.877 | Aggregate |
| Mistral Misconception | Misconception Contrast | 0.751 | 0.912 | 0.912 | Aggregate |
| Mistral Probe LOF Mean | Probe | 0.936 | 0.978 | 0.928 | 28 |
| Mistral Probe Mahalanobis Mean | Probe | 0.932 | 0.948 | 0.942 | 28 |
| Mistral Rephrase | Iterative Rephrase | 0.681 | 0.681 | 0.681 | Aggregate |

Table 15: Online anomaly detection results for the Population dataset. *LOF*: local outlier factor *Mean AUROC*: AUROC of per-layer anomaly scores, averaged, *Agg. AUROC*: AUROC of anomaly scores aggregated across layers, *Best AUROC*: highest AUROC achieved in any layer or aggregated across layers.

## B.7 SciQ Results

| Score | Features | Mean AUROC | Agg. AUROC | Best AUROC | Best Layer |
|-------|----------|------------|------------|------------|------------|
| Meta Activations QUE | Activations | 0.443 | 0.386 | 0.386 | Aggregate |
| Mistral Activations QUE | Activations | **0.481** | **0.637** | **0.637** | Aggregate |
| Mistral GMM | Activations | 0.463 | - | 0.396 | 19 |
| Mistral Likelihood Ratio | Activations | 0.465 | - | 0.396 | 19 |

Table 16: Offline anomaly detection results for the SciQ dataset. *QUE*: quantum entropy, *GMM*: Gaussian Mixture Model, *Mean AUROC* AUROC of per-layer anomaly scores, averaged, *Agg. AUROC*: AUROC of anomaly scores aggregated across layers, *Best AUROC*: highest AUROC achieved in any layer or aggregated across layers.

| Score | Features | Mean AUROC | Agg. AUROC | Best AUROC | Best Layer |
|---|---|---|---|---|---|
| Meta Activations LOF | Activations | 0.572 | 0.572 | 0.572 | Aggregate |
| Meta Activations Mahalanobis | Activations | 0.419 | 0.365 | 0.517 | 10 |
| Meta Attribution LOF Mean | Attribution | 0.434 | 0.410 | 0.519 | 16 |
| Meta Attribution Mahalanobis Mean | Attribution | 0.380 | 0.383 | 0.384 | 31 |
| Meta Flow Laplace | NFlow | 0.462 | 0.457 | 0.474 | 23 |
| Meta Flow Mahalanobis | NFlow | 0.344 | 0.340 | 0.340 | Aggregate |
| Meta Probe LOF Mean | Probe | 0.403 | 0.362 | 0.427 | 28 |
| Meta Probe Mahalanobis Mean | Probe | 0.367 | - | 0.359 | 22 |
| Meta SAE Diag Mahalanobis | SAE | 0.388 | 0.393 | 0.393 | 23 |
| Meta SAE $L_0$ | SAE | 0.471 | 0.455 | 0.486 | 23 |
| Mistral Activations LOF | Activations | 0.547 | 0.619 | 0.619 | Aggregate |
| Mistral Activations Mahalanobis | Activations | 0.464 | - | 0.384 | 25 |
| Mistral Attribution LOF Mean | Attribution | **0.575** | **0.642** | **0.642** | Aggregate |
| Mistral Attribution Mahalanobis Mean | Attribution | 0.491 | 0.600 | 0.600 | Aggregate |
| Mistral Misconception | Misconception Contrast | 0.571 | 0.605 | 0.605 | Aggregate |
| Mistral Probe LOF Mean | Probe | 0.460 | 0.457 | 0.579 | 28 |
| Mistral Probe Mahalanobis Mean | Probe | 0.494 | 0.494 | 0.548 | 28 |
| Mistral Rephrase | Iterative Rephrase | 0.405 | 0.405 | 0.405 | Aggregate |

Table 17: Online anomaly detection results for the SciQ dataset. *LOF*: local outlier factor *Mean AUROC*: AUROC of per-layer anomaly scores, averaged, *Agg. AUROC*: AUROC of anomaly scores aggregated across layers, *Best AUROC*: highest AUROC achieved in any layer or aggregated across layers.

## B.8 SENTIMENT RESULTS

| Score | Features | Mean AUROC | Agg. AUROC | Best AUROC | Best Layer |
|---|---|---|---|---|---|
| Meta Activations QUE | Activations | 0.635 | 0.789 | 0.789 | Aggregate |
| Mistral Activations QUE | Activations | **0.746** | **0.975** | **0.975** | Aggregate |
| Mistral GMM | Activations | 0.636 | - | 0.790 | 19 |
| Mistral Likelihood Ratio | Activations | 0.642 | - | 0.806 | 19 |

Table 18: Offline anomaly detection results for the Sentiment dataset. *QUE*: quantum entropy, *GMM*: Gaussian Mixture Model, *Mean AUROC* AUROC of per-layer anomaly scores, averaged, *Agg. AUROC*: AUROC of anomaly scores aggregated across layers, *Best AUROC*: highest AUROC achieved in any layer or aggregated across layers.

| Score | Features | Mean AUROC | Agg. AUROC | Best AUROC | Best Layer |
|---|---|---|---|---|---|
| Meta Activations LOF | Activations | 0.488 | 0.488 | 0.488 | Aggregate |
| Meta Activations Mahalanobis | Activations | 0.532 | 0.541 | 0.503 | 10 |
| Meta Attribution LOF Mean | Attribution | 0.622 | 0.659 | 0.687 | 16 |
| Meta Attribution Mahalanobis Mean | Attribution | 0.441 | 0.322 | **0.809** | 31 |
| Meta Flow Laplace | NFlow | 0.547 | 0.565 | 0.574 | 23 |
| Meta Flow Mahalanobis | NFlow | 0.548 | 0.553 | 0.553 | Aggregate |
| Meta Probe LOF Mean | Probe | 0.683 | 0.741 | 0.795 | 28 |
| Meta Probe Mahalanobis Mean | Probe | 0.501 | - | 0.637 | 22 |
| Meta SAE Diag Mahalanobis | SAE | 0.460 | 0.432 | 0.514 | 23 |
| Meta SAE $L_0$ | SAE | 0.468 | 0.454 | 0.493 | 23 |
| Mistral Activations LOF | Activations | 0.619 | 0.658 | 0.658 | Aggregate |
| Mistral Activations Mahalanobis | Activations | 0.597 | - | 0.718 | 25 |
| Mistral Attribution LOF Mean | Attribution | 0.480 | 0.432 | 0.432 | Aggregate |
| Mistral Attribution Mahalanobis Mean | Attribution | 0.424 | 0.404 | 0.404 | Aggregate |
| Mistral Misconception | Misconception Contrast | 0.526 | 0.523 | 0.523 | Aggregate |
| Mistral Probe LOF Mean | Probe | 0.419 | 0.409 | 0.480 | 28 |
| Mistral Probe Mahalanobis Mean | Probe | 0.421 | 0.417 | 0.468 | 28 |
| Mistral Rephrase | Iterative Rephrase | 0.298 | 0.298 | 0.298 | Aggregate |

Table 19: Online anomaly detection results for the Sentiment dataset. *LOF*: local outlier factor *Mean AUROC*: AUROC of per-layer anomaly scores, averaged, *Agg. AUROC*: AUROC of anomaly scores aggregated across layers, *Best AUROC*: highest AUROC achieved in any layer or aggregated across layers.

### B.9 SQUARING RESULTS

| Score | Features | Mean AUROC | Agg. AUROC | Best AUROC | Best Layer |
|---|---|---|---|---|---|
| Meta Activations QUE | Activations | **1.000** | **1.000** | **1.000** | Aggregate |

Table 20: Offline anomaly detection results for the Squaring dataset. *QUE*: quantum entropy, *Mean AUROC* AUROC of per-layer anomaly scores, averaged, *Agg. AUROC*: AUROC of anomaly scores aggregated across layers, *Best AUROC*: highest AUROC achieved in any layer or aggregated across layers.

| Score | Features | Mean AUROC | Agg. AUROC | Best AUROC | Best Layer |
|---|---|---|---|---|---|
| Meta Activations LOF | Activations | 0.915 | 0.915 | 0.915 | Aggregate |
| Meta Activations Mahalanobis | Activations | 0.930 | **1.000** | 0.915 | 10 |
| Meta Attribution LOF Mean | Attribution | 0.951 | 0.941 | 0.956 | 16 |
| Meta Attribution Mahalanobis Mean | Attribution | 0.922 | 0.949 | 0.995 | 31 |
| Meta Flow Laplace | NFlow | 0.999 | **1.000** | **1.000** | 23 |
| Meta Flow Mahalanobis | NFlow | **1.000** | **1.000** | **1.000** | Aggregate |
| Meta Probe LOF Mean | Probe | 0.925 | 0.930 | 0.936 | 28 |
| Meta Probe Mahalanobis Mean | Probe | 0.917 | - | 0.930 | 22 |
| Meta SAE Diag Mahalanobis | SAE | 0.992 | **1.000** | 0.977 | 23 |
| Meta SAE $L_0$ | SAE | **1.000** | **1.000** | **1.000** | 23 |
| Mistral Attribution LOF Mean | Attribution | 0.931 | 0.966 | 0.966 | Aggregate |
| Mistral Attribution Mahalanobis Mean | Attribution | 0.891 | 0.942 | 0.942 | Aggregate |
| Mistral Probe LOF Mean | Probe | 0.953 | 0.976 | 0.996 | 28 |
| Mistral Probe Mahalanobis Mean | Probe | 0.905 | 0.965 | 0.994 | 28 |

Table 21: Online anomaly detection results for the Squaring dataset. *LOF*: local outlier factor *Mean AUROC*: AUROC of per-layer anomaly scores, averaged, *Agg. AUROC*: AUROC of anomaly scores aggregated across layers, *Best AUROC*: highest AUROC achieved in any layer or aggregated across layers.

## B.10 SUBTRACTION RESULTS

| Score | Features | Mean AUROC | Agg. AUROC | Best AUROC | Best Layer |
|---|---|---|---|---|---|
| Meta Activations QUE | Activations | **0.918** | **1.000** | **1.000** | Aggregate |
| Mistral Activations QUE | Activations | 0.915 | **1.000** | **1.000** | Aggregate |
| Mistral GMM | Activations | 0.898 | - | **1.000** | 19 |
| Mistral Likelihood Ratio | Activations | 0.902 | - | **1.000** | 19 |

Table 22: Offline anomaly detection results for the Subtraction dataset. *QUE*: quantum entropy, *GMM*: Gaussian Mixture Model, *Mean AUROC* AUROC of per-layer anomaly scores, averaged, *Agg. AUROC*: AUROC of anomaly scores aggregated across layers, *Best AUROC*: highest AUROC achieved in any layer or aggregated across layers.

| Score | Features | Mean AUROC | Agg. AUROC | Best AUROC | Best Layer |
|---|---|---|---|---|---|
| Meta Activations LOF | Activations | **1.000** | **1.000** | **1.000** | Aggregate |
| Meta Activations Mahalanobis | Activations | 0.907 | **1.000** | **1.000** | 10 |
| Meta Attribution LOF Mean | Attribution | 0.868 | 0.931 | 0.856 | 16 |
| Meta Attribution Mahalanobis Mean | Attribution | 0.750 | 0.812 | 0.848 | 31 |
| Meta Flow Laplace | NFlow | 0.507 | 0.426 | 0.906 | 23 |
| Meta Flow Mahalanobis | NFlow | 0.998 | 1.000 | 1.000 | Aggregate |
| Meta Probe LOF Mean | Probe | 0.859 | 0.915 | 0.932 | 28 |
| Meta Probe Mahalanobis Mean | Probe | 0.795 | - | 0.806 | 22 |
| Meta SAE Diag Mahalanobis | SAE | 0.997 | 1.000 | **1.000** | 23 |
| Meta SAE $L_0$ | SAE | 0.993 | 1.000 | 1.000 | 23 |
| Mistral Activations LOF | Activations | 0.889 | 0.999 | 0.999 | Aggregate |
| Mistral Activations Mahalanobis | Activations | 0.899 | - | **1.000** | 25 |
| Mistral Attribution LOF Mean | Attribution | 0.853 | 0.974 | 0.974 | Aggregate |
| Mistral Attribution Mahalanobis Mean | Attribution | 0.951 | 0.970 | 0.970 | Aggregate |
| Mistral Misconception Contrast | Misconception | 0.411 | - | - | Aggregate |
| Mistral Probe LOF Mean | Probe | 0.928 | 0.998 | 0.994 | 28 |
| Mistral Probe Mahalanobis Mean | Probe | 0.886 | 0.998 | 0.993 | 28 |
| Mistral Rephrase | Iterative Rephrase | 0.496 | 0.496 | 0.496 | Aggregate |

Table 23: Online anomaly detection results for the Subtraction dataset. *LOF*: local outlier factor *Mean AUROC*: AUROC of per-layer anomaly scores, averaged, *Agg. AUROC*: AUROC of anomaly scores aggregated across layers, *Best AUROC*: highest AUROC achieved in any layer or aggregated across layers.

## B.11    LAYERWISE ANOMALY DETECTOR RESULTS

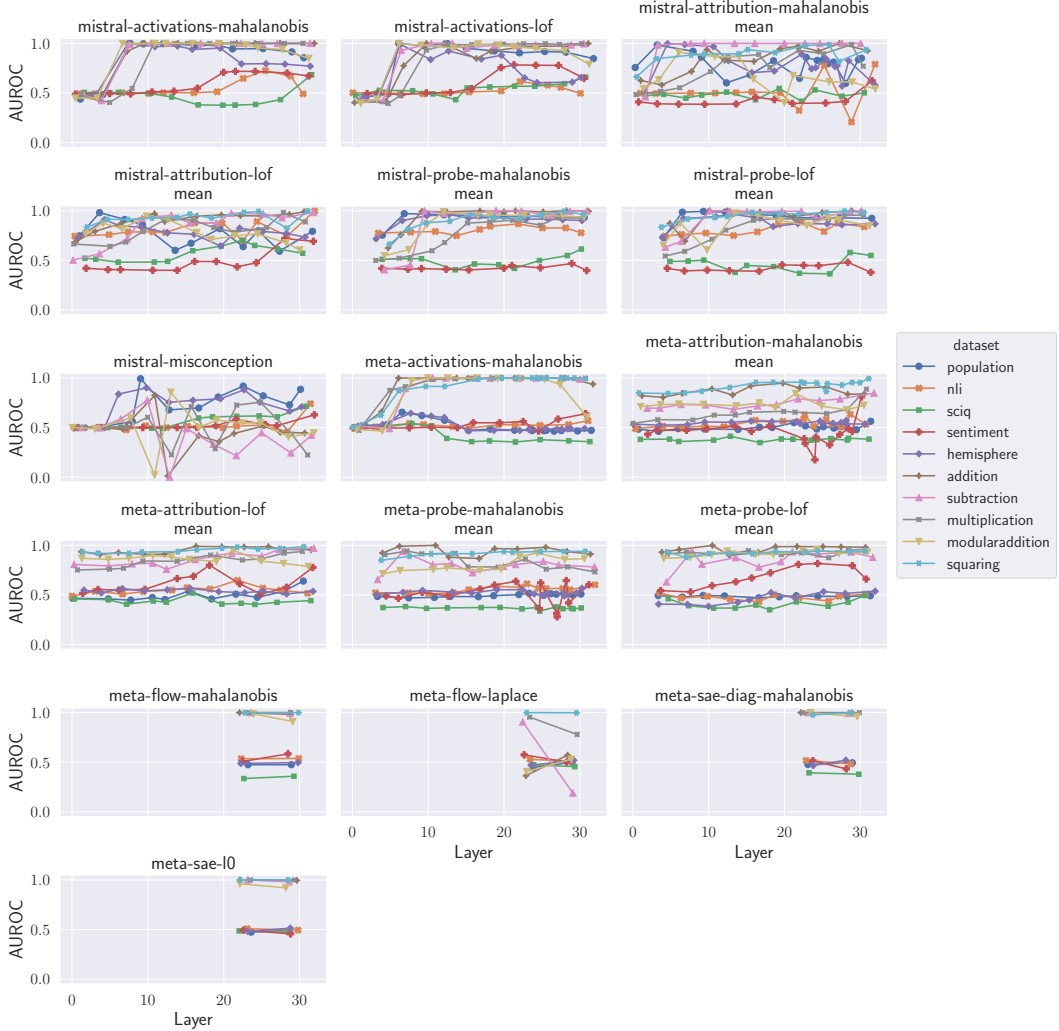

Figure 6: Layerwise AUC performance of online anomaly detectors across all datasets. Similar to the offline case, the x-axis represents the layer index, and the y-axis shows the AUC scores. The results indicate that certain anomaly detection methods, such as meta-activations-mahalanobis and meta-probe-lof mean, demonstrate strong performance at deeper layers, highlighting the importance of feature extraction depth in anomaly detection.

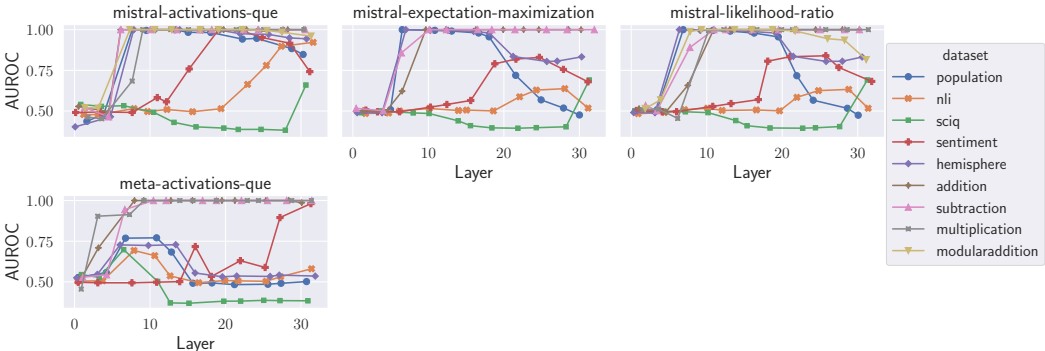

Figure 7: Layerwise AUC performance of offline anomaly detectors across all datasets. The x-axis represents the layer index, while the y-axis indicates the AUC scores for distinguishing between normal and anomalous inputs. We observe that some methods, such as mistral-activations-que, maintain high AUC scores across layers, whereas others exhibit variability depending on the dataset.

