# OpenReview forum: "Mechanistic Anomaly Detection for "Quirky'' Language Models"
_ICLR.cc/2025/Workshop/BuildingTrust — BuildingTrust_

### Official Review · Reviewer_QSF5 · 2025-02-26
**It is a great workshop paper on exploring ways to build detectors based on internal features**

**Rating:** 7
**Confidence:** 2

**Review:**

Overall, I think it's an excellent workshop paper because:
- This area requires significant effort, and the paper comprehensively explores how to build such detectors and evaluate their effectiveness. Although the findings are not consistent, the paper still makes a good contribution to this promising field at the workshop level.
- The methodology used—fine-tuning models to behave normally versus anomalously when triggered by specific names—is an interesting approach to studying this problem.

Some minor issues:
- There are several typos (e.g., lines 18, 58, 73, etc.).
- The auto-reference on line 324 is unclear.

---

### Official Review · Reviewer_jQit · 2025-02-28

**Rating:** 6
**Confidence:** 4

**Review:**

The paper addresses the increasing difficulty of supervising LLMs by introducing Mechanistic Anomaly Detection (MAD), a method that uses model's hidden representations to flag anomalous training signals.

Pros
1. The use of 'quirky' datasets inspired by Mallen et al. (2024) ensures a controlled testing environment where models exhibit systematic biases. The variety of tasks strengths the validity of the experiments

Cons:
1. The study does not assess MAD's effectiveness on real-world LLM deployments, where anomalies might be more nuanced and context dependent.
2. The study focuses on models deliberately trained to exhibit anomalous behavior, raising the question of wether the findings generalize to naturally occurring anomalies in frontier models.

---

### Official Review · Reviewer_MARM · 2025-03-02

**Rating:** 6
**Confidence:** 3

**Review:**

This paper claims to measure anomalous behavior of models. Anomaly detectors are trained for the same.

The paper defines their notion of anomaly, which they call quirkiness. I have doubts if this can actually be called anomaly and with the setting of the paper. Does anomalous behavior have to emerge out of some sort of training? Can anomalies not emerge due to OOD test data? Even though the authors test on harder samples which can be considered OOD, they still finetune the model on easier data leading to some sort of bias being introduced. Also there can be different behavior when considering pretraining vs. finetuning.

---

### Decision · Program_Chairs · 2025-03-04

**Decision:**

Accept

**Comment:**

The paper's definition of "quirkiness" as an anomaly is questionable, as it does not fully consider naturally occurring anomalies, such as those arising from out-of-distribution (OOD) test data. Additionally, the study's focus on models deliberately trained to exhibit anomalous behavior raises concerns about the generalizability of its findings to real-world LLM deployments, where anomalies may be more nuanced and context-dependent.